# Molecular surveillance of resistance to pyrethroids insecticides in Colombian *Aedes aegypti* populations

**Yurany Granada, Ana María Mejía-Jaramillo [ID], Sara Zuluaga [ID], Omar Triana-Chávez [ID]** *

Grupo Biología y Control de Enfermedades Infecciosas, Universidad de Antioquia UdeA, Medellin, Colombia

* omar.triana@udea.edu.co

## Abstract

### Introduction

In Colombia, organochloride, organophosphate, carbamate, and pyrethroid insecticides are broadly used to control *Aedes aegypti* populations. However, Colombian mosquito populations have shown variability in their susceptibility profiles to these insecticides, with some expressing high resistance levels.

### Materials and methods

In this study, we analyzed the susceptibility status of ten Colombian field populations of *Ae. aegypti* to two pyrethroids; permethrin (type-I pyrethroid) and lambda-cyhalothrin (type-II pyrethroid). In addition, we evaluated if mosquitoes pressured with increasing lambda-cyhalothrin concentrations during some filial generations exhibited altered allelic frequency of these *kdr* mutations and the activity levels of some metabolic enzymes.

### Results

Mosquitoes from all field populations showed resistance to lambda-cyhalothrin and permethrin. We found that resistance profiles could only be partially explained by *kdr* mutations and altered enzymatic activities such as esterases and mixed-function oxidases, indicating that other yet unknown mechanisms could be involved. The molecular and biochemical analyses of the most pyrethroid-resistant mosquito population (Acacías) indicated that *kdr* mutations and altered metabolic enzyme activity are involved in the resistance phenotype expression.

### Conclusions

In this context, we propose genetic surveillance of the mosquito populations to monitor the emergence of resistance as an excellent initiative to improve mosquito-borne disease control measures.

**Data Availability Statement:** All relevant data are within the manuscript and its Supporting Information files.

**Funding:** OTC was funded by CODI - Universidad de Antioquia, UdeA, Grant CPT-2005 and British Council Institutional Links Newton Fund. YG has a fellowship from Gobernación del Tolima and Universidad del Tolima (Project BPIN 2013000100103). The Funders had no role in study design, data collection and analysis, decision to publish, or preparation of the manuscript.

## Author summary

The main method of preventing *Aedes*-borne diseases such as dengue, Zika, and chikungunya is by targeting the primary mosquito vector, *Aedes aegypti*, with insecticides. However, the success of these vector control strategies is jeopardized by the widespread development of insecticide resistance in mosquito populations. Furthermore, the molecular mechanisms of insecticide resistance in *Ae. aegypti* are still not well understood, resulting in limited resistance mitigation and management strategies. In this paper, we found that resistance to some pyrethroid insecticides in different Colombian cities is associated with three allelic substitutions V419L, V1016I, and F1534C, on the voltage-gated sodium channel gene, known as *kdr* ('knock-down resistance') mutations, with all three mutations present in mosquitoes resistant to pyrethroids. The data also showed that *kdr* mutations are important in conferring low resistance levels, but after around 10-fold intensity, the allele frequencies don't change, indicating that other mechanisms contribute to the resistance. Thus, we found that mosquitoes under selective pressure with insecticides present also altered enzymatic activities such as esterases and mixed-function oxidases, indicating that *kdr* mutations and metabolic enzymes are involved in the resistance expression. The findings on the extent of insecticide resistance and the molecular mechanisms underpinning the problem will impact the surveillance, selection, and rational use of insecticides by local health authorities.

## 1. Introduction

The World Health Organization has listed dengue as one of its top ten global health priorities, with several countries facing unprecedented outbreaks and declaring states of emergency, and 2019 was a record year in terms of infections and outbreaks [1]. So far, in 2020 and 2021, the number of dengue cases has increased dramatically in different Latin American countries, which the COVID-19 pandemic has further exacerbated.

Insecticides are widely used to control mosquitoes and other insect vectors that transmit different pathogens. Types of insecticides include ovicide, larvicide, and adulticide, used against insect eggs, larvae, and adults, respectively. Nearly all insecticides are toxic to humans and/or mammals, but pyrethroids are the best option, as they are less harmful than carbamates and organophosphates. Pyrethroids interrupt the mosquito's nerve function by binding to the voltage-dependent sodium channel proteins, causing repetitive discharges, depolarizing the axonal membrane, and causing synaptic abnormalities that generate paralysis and, ultimately, death of the mosquito [2]. There are two types of pyrethroids; those lacking the α-cyano group, classified as type I pyrethroids, and those with the α-cyano called type II pyrethroids.

The lack of sensitivity of *Ae. aegypti* to pyrethroids is associated with higher activity and expression of the insect's detoxifying enzymes, including unspecified esterases, glutathione S-transferases, and mixed-function oxidases [3–7]. Insensitivity is also attributed to increased expression of cuticular genes [8,9] and the mosquito's microbiota [10–12]. However, the primary mechanism contributing to the phenomena are mutations in the voltage-gated sodium channel coding gene, commonly called knockdown resistance (*kdr*) [13]. Although resistance to pyrethroids has been progressively documented in insects, the biochemical and molecular mechanisms by which *Ae. aegypti* develops pyrethroid insecticide resistance are not fully understood, making it challenging to design or improve effective mosquito control methods in different regions around the world [14–17].

In Colombia, organochlorides, organophosphates, carbamates, and pyrethroids have been broadly used to control *Ae. aegypti* populations. However, these mosquito populations have

shown variability in susceptibility profiles to these insecticides, with some of them expressing high resistance levels. Previous studies have determined that *Ae. aegypti* populations gain resistance to lambda-cyhalothrin through point mutations in the voltage-gated sodium channel gene [18–20]. Furthermore, a positive association between the V410L and V1016I mutations, but not the F1534C mutation, and resistance to this insecticide have been reported, suggesting that F1534C mutation could be conferring resistance to other insecticides [13,18,21–23].

Moreover, metabolic resistance due to increased activity levels of key insecticide-degrading enzymes has been described in *Aedes* populations from different regions of Colombia [19,20]. However, the role of these mechanisms when the mosquitoes are exposed to high concentrations of insecticides is not well understood. In the present study, we analyzed the susceptibility status of ten Colombian field populations of *Ae. aegypti* to permethrin (type-I pyrethroid) and lambda-cyhalothrin (type-II pyrethroid). We also determined if increasing the lambda-cyhalothrin concentration over multiple generations could alter the allelic frequency of these *kdr* mutations and the activity levels of some metabolic enzymes. Our results indicated that there is a generalized resistance to pyrethroids throughout the country, mainly to type I pyrethroids. Also, we found that the resistance to pyrethroids in Colombian mosquito populations is associated with three point mutations (V419L, V1016I, and F1534C) on the voltage-gated sodium channel gene. However, we showed that the resistance profile could only be partially explained by these *kdr* mutations and altered enzymatic activities such as the esterases and mixed-function oxidases, indicating other mechanisms yet unknown could be involved. In this context, we propose genetic surveillance of the mosquito populations to monitor the emergence of insecticide resistance, which could potentially improve mosquito control measures.

## 2. Results

### 2.1. Mosquitoes from ten different regional populations were resistant to permethrin and lambda-cyhalothrin pyrethroids

A total of 32,760 *Ae. aegypti* mosquitoes were tested for susceptibility to permethrin and lambda-cyhalothrin insecticides. Resistance to permethrin was observed in all ten mosquito populations tested. The most resistant population to permethrin was Cúcuta, with an $RR_{50}$ of 152; the least resistant populations were Honda ($RR_{50}$ of 13.82), and Itagüí and Neiva (Both with $RR_{50}$ 18.43; Fig 1 and Table 1).

In addition, four of the ten populations evaluated (Bello, Itagüí, Moniquirá, and Puerto Bogotá) showed moderate resistance to lambda-cyhalothrin insecticide with $RR_{50}$ of 7.03, 7.8, 8.7, and 9.1, respectively. The field-collected Acacías population was the most resistant, with an $RR_{50}$ of 31.64 (Fig 1 and Table 1). Overall, mosquitoes from all localities showed lower $RR_{50}$ values to lambda-cyhalothrin than to permethrin insecticide (Fig 2, top).

Remarkably, the Acacías mosquito population exposed to lambda-cyhalothrin over six generations showed an $RR_{50}$ of 1,608 to permethrin and 694 to lambda-cyhalothrin, increasing its resistance status 32 and 22 times, respectively, compared with the field-collected parental population. In contrast, the Acacías population without insecticide pressure showed a 1.7-fold increase in the $RR_{50}$ to permethrin and a 1.7-fold decrease in the $RR_{50}$ to lambda-cyhalothrin compared to the parental population (Table 1).

### 2.2. The kdr mutations are present in all Colombian Ae. aegypti populations studied

After analyzing at least 552 mosquitoes, the allele-specific PCR (AS-PCR) assay revealed the high frequency of the three *kdr* mutations, V410L, V1016I, and F1534C, in the field-caught

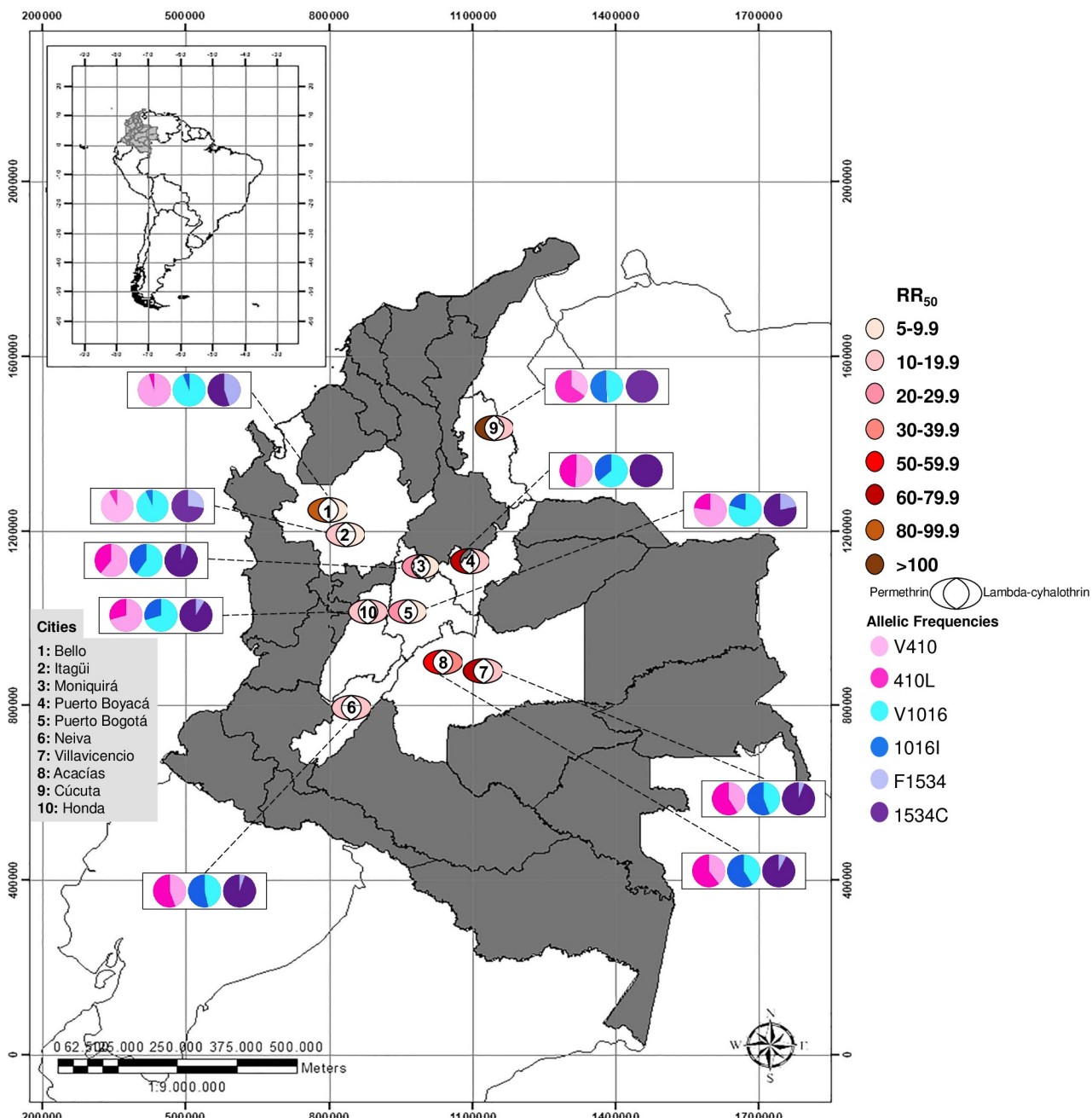

**Fig 1. Colombian cities in which *Aedes aegypti* mosquitoes were collected.** Numbers correspond to each city: 1. Bello, 2. Itagüí, 3. Moniquirá, 4. Puerto Boyacá, 5. Puerto Bogotá, 6. Neiva, 7. Villavicencio, 8. Acacías, 9. Cúcuta, and 10. Honda. For each city, the level of resistance ($RR_{50}$) to permethrin (to the left of the number) and lambda-cyhalothrin (to the right of the number) in comparison to the susceptible Rockefeller strain of mosquito is shown in the half-round; darker colors represent a higher degree of insecticide resistance. The distribution of the *kdr* alleles for the positions V410L (pink), V1016I (blue), and F1534C (purple) are shown for each city. Lighter colors represent the wt allele frequencies, and darker colors indicate the mutated allele frequencies. Shaped downloaded from http://tapiquen-sig.jimdofree.com., Carlos Efraín Porto Tapiquén. Geografía, SIG y Cartografía Digital. Valencia, Spain, 2020 (free distribution).

Colombian *Ae. aegypti* populations (Figs 1 and 2). In the populations that were more susceptible to lambda-cyhalothrin (Bello and Itagüí), the frequency of the 410L mutant allele ranged between 0.05–0.08, while the most resistant populations showed allele frequencies ranging

**Table 1. Resistance ratio to Permethrin and Lambda-cyhalothrin of *Aedes aegypti* populations studied in Colombia.**

| *Ae. aegypti* population | N | Permethrin | | | | Lambda-cyhalothrin | | | |
|---|---|---|---|---|---|---|---|---|---|
| | | LC50 (95% CI) | LC90 (95% CI) | RR50 | RR90 | LC50 (95% CI) | LC90 (95% CI) | RR50 | RR90 |
| Rockefeller | 1260 | 0.000217(0.000192–0.000244) | 0.000796(0.000670–0.000821) | 1 | 1 | 0.000474(0.000398–0.000564) | 0.0016(0.0012–0.0022) | 1 | 1 |
| Bello (1) | 1260 | 0.021(0.006–0.045) | 0.048(0.032–0.144) | 96.77** | 221.19 | 0.0033(0.0023–0.0046) | 0.010(0.008–0.015) | 7.03* | 6.25 |
| Itagüí (2) | 1260 | 0.004(0.003–0.004) | 0.008(0.007–0.010) | 18.43** | 10.05 | 0.0037(0.0032–0.0042) | 0.0084(0.0070–0.011) | 7.8* | 5.25 |
| Moniquirá (3) | 1260 | 0.006(0.005–0.007) | 0.010(0.009–0.012) | 27.64** | 12.56 | 0.004124(0.002–0.006) | 0.013 (0.012–0.016) | 8.7* | 8.12 |
| Puerto Boyacá (4) | 1260 | 0.015(0.001–0.026) | 0.046(0.033–0.093) | 69.12** | 57.79 | 0.006(0.005–0.007) | 0.019 (0.016–0.025) | 12.66** | 11.87 |
| Puerto Bogotá (5) | 1260 | 0.006(0.005–0.006) | 0.014(0.012–0.016) | 27.65** | 17.59 | 0.00433(0.003–0.005) | 0.012(0.009–0.018) | 9.1* | 7.5 |
| Neiva (6) | 1260 | 0.004(0.002–0.006) | 0.009(0.007–0.012) | 18.43** | 11.31 | 0.005(0.004–0.006) | 0.0105(0.00805–0.0145) | 10.55** | 6.56 |
| Villavicencio (7) | 1260 | 0.015(0.014–0.016) | 0.041(0.036–0.049) | 69.12** | 51.51 | 0.008(0.006–0.0096) | 0.024 (0.019–0.031) | 16.87** | 15 |
| Acacías (8) | 1260 | 0.011(0.010–0.012) | 0.040(0.034–0.0489) | 50.69** | 50.25 | 0.015(0.012–0.018) | 0.05 (0.04–0.09) | 31.64** | 31.25 |
| Cúcuta (9) | 1260 | 0.033(0.027–0.040) | 0.135(0.98–0.210) | 152.07** | 169.58 | 0.009(0.002–0.10) | 0.016(0.010–0.34) | 18.98** | 10 |
| Honda (10) | 1260 | 0.003(0.003–0.004) | 0.009(0.008–0.011) | 13.82** | 11.31 | 0.0056(0.00492–0.0063) | 0.0174(0.0153–0.020) | 11.81** | 10.87 |
| Acacías F7 pressure | 1260 | 0.349(0.21–0.50) | 1.127(0.847–1.834) | 1608.29** | 1415.82 | 0.329(0.252–0.430) | 0.798(0.644–1.080) | 694.09** | 498.75 |
| Acacías F7 without pressure | 1260 | 0.019(0.016–0.023) | 0.036(0.031–0.045) | 87.55** | 45.22 | 0.009(0.007–0.010) | 0.033(0.026–0.044) | 18.99** | 20.62 |
| **TOTAL** | **16380** | | | | | | | | |

* Tolerant population

** Resistant population.

from 0.61–0.65 (Acacías and Cúcuta; Figs 1 and 2). Similarly, the V1016I mutant allele frequency changed from 0.06 in susceptible mosquito populations (Bello) to 0.51–0.59 in the more resistant mosquitoes (Cúcuta and Acacías). In contrast, the 1534C mutation was found in high frequencies in all the populations, with allele frequencies upwards of 0.55. Remarkably, every individual collected from Puerto Boyacá and Cúcuta presented this fixed mutated allele (i.e., frequency of 1). The control Rockefeller strain did not show the mutated allele in any of the cases.

For Loci 410 and 1016, all genotypes were found to be in Hardy-Weinberg equilibrium, except Puerto Bogotá and Cúcuta, and Puerto Boyacá and Cúcuta, for locus 410 and locus 1016, respectively. Finally, for locus 1534, Itagüí and Villavicencio populations were not in Hardy-Weinberg equilibrium (p$\leq$ 0.05; S1 Table).

Regarding the inbreeding coefficients ($F_{IS}$) for locus 410, a deficiency of heterozygotes was notorious in mosquito populations from Puerto Bogotá and Cúcuta where $F_{IS}$ values $> 0$ were observed. In contrast, for locus 1016, most of the populations presented an excess of heterozygotes (value $< 0$), with Puerto Boyacá and Cúcuta recording the highest values. Finally, for locus 1534, an excess of heterozygotes was observed in mosquitoes from Itagüí and a deficit in mosquitoes from Villavicencio (S1 Table).

Additionally, the dynamics of allele frequencies were analyzed for the Bello and Villavicencio localities, where we collected samples at different times across four years (2012–2016). In

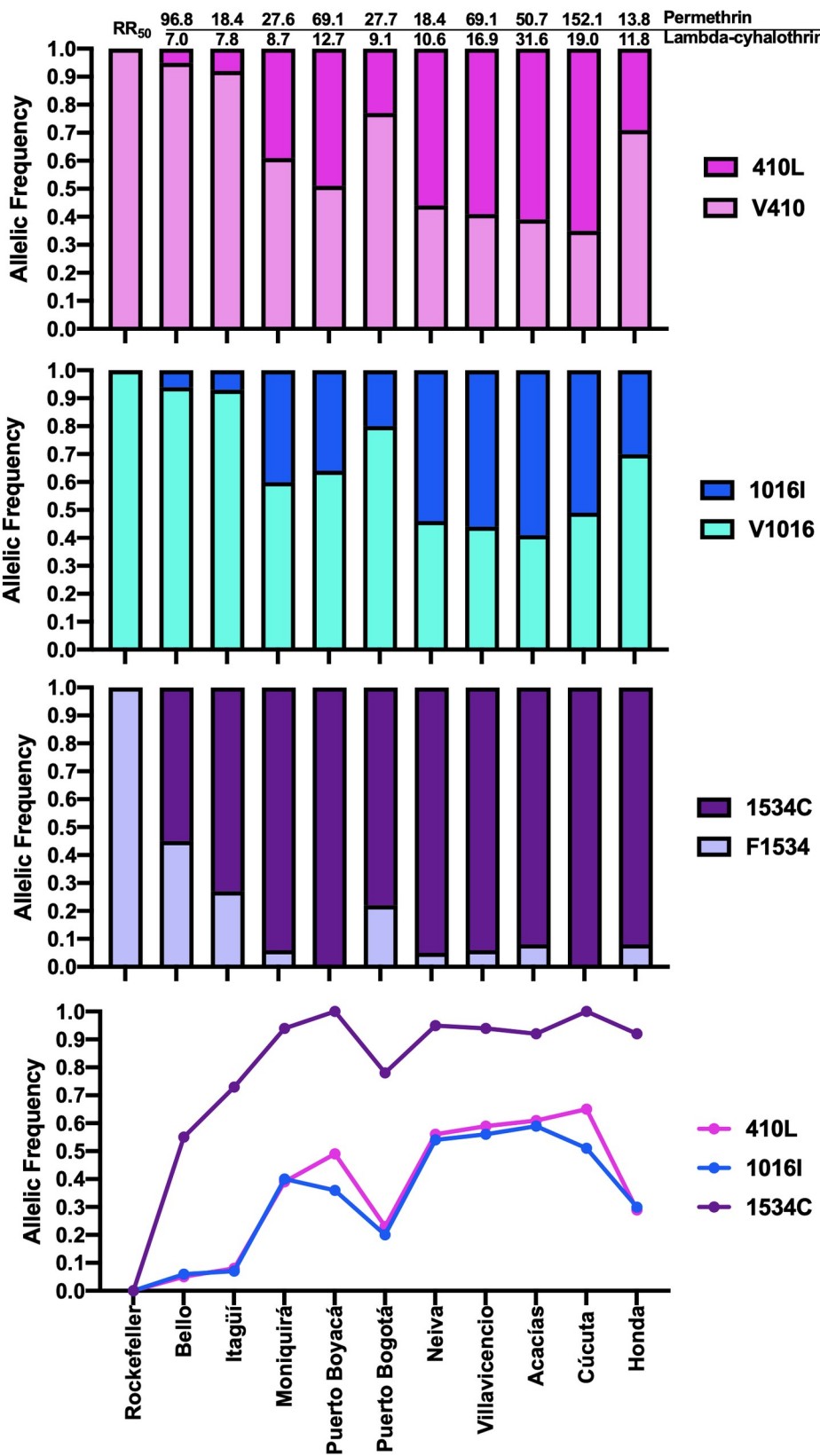

**Fig 2. Allelic frequencies for the positions V410L (pink), V1016I (blue), and F1534C (purple) in Colombian *Aedes aegypti* study populations.** The light colors in the bars indicate the wt allele frequencies; the dark colors indicate the mutated allele frequencies. The RR value for mosquitoes from each city is shown above the bars in the top graph. The bottom chart shows the variation of mutated allele frequencies for the three positions in each city.

2012, mosquitoes from the Bello municipality were reported as susceptible to lambda-cyhalothrin, and they presented frequencies of the 410L, 1016I, and 1534C resistant alleles of 0.05, 0.04, and 0.56, respectively [18]. In 2016 (this study), the population was described as moderately resistant, and the mutant alleles showed a slight increase, with frequencies of 0.06, 0.06, and 0.55. From 2012 through 2016, the percentage of resistant alleles in mosquitoes from Villavicencio increased by 0.45, 0.41, and 0.91 to 0.59, 0.56, and 0.94 in 410L, 1016I, and 1534C mutations, respectively (Fig 3A and 3B). Interestingly, the Villavicencio population from 2016 presented a deficiency of heterozygotes, and it was in Hardy-Weinberg disequilibrium (S1 Table).

## 2.3. Mosquitoes under insecticide pressure change the resistance status but not the allelic frequencies

To determine the allelic frequency dynamic in the absence and presence of lambda-cyhalothrin insecticide, we submitted mosquitoes from the Acacías population to continuous pressure with the $LC_{50}$ and $LC_{90}$ of the insecticide during six generations. The allelic frequencies of 410L and 1016I *kdr* mutations decreased significantly when mosquitoes were kept without insecticide pressure (Fig 3C). However, after pressure with the lambda-cyhalothrin insecticide, resistance to permethrin and lambda-cyhalothrin increased 32 and 22 times, respectively (Table 1), but the allelic frequencies of the mutated alleles did not change significantly (0.61 to 0.58, 0.59 to 0.41, and 0.92 to 1, for 410L, 1016I, and 1534C, respectively; Fig 3C). However, for the 1016 allele, a Hardy-Weinberg disequilibrium and an excess of heterozygotes ($F_{IS}$ values $< 0$) were observed in comparison with the field population (S1 Table).

## 2.4. Mutations in combination

The three-loci genotypes were thoroughly analyzed in 482 mosquitoes using PCR. After analyzing all the possible combinations of the wild-type and mutated alleles for the three positions, only 14 out of 27 of the possible genotypes were present in the mosquito populations studied (Fig 4). However, only four combinations were present in 82.57% (398/482) of individuals analyzed. The most frequent genotype (29.25%, 141/482 mosquitoes) was the double heterozygous for loci 410 and 1016 and homozygous resistant for locus 1534 ($VL_{410}/VI_{1016}/CC_{1534}$). The second most frequent genotype was mosquitoes carrying only the mutation 1534C and double homozygous wild-type genotype ($VV_{410}/VV_{1016}/CC_{1534}$) for loci 410 and 1016 (23%, 111/482). The third most common genotype was the F1534C heterozygous genotype ($VV_{410}/VV_{1016}/FC_{1534}$), found at a frequency of 16.8% (81/482). Finally, the triple homozygous mutant ($LL_{410}/II_{1016}/CC_{1534}$) was present in 13.48% (65/482) of mosquitoes from Moniquirá, Puerto Bogotá, Neiva, Villavicencio, Acacías and Cúcuta. Interestingly, the triple wild-type homozygous ($VV_{410}/VV_{1016}/FF_{1534}$) was present in just 2.69% (13/482) of mosquitoes collected from Bello, Itagüí, and Puerto Bogotá, which were the populations more susceptible to insecticides. Remarkably, the 410L and 1016I *kdr* mutations were never found alone; they were always accompanied by other mutations.

On the other hand, the Acacías populations, with or without insecticide pressure, showed the loss of some allele combinations compared with the parental population. Thus, there were no mosquitoes in both populations with the double homozygous wild-type genotype for loci

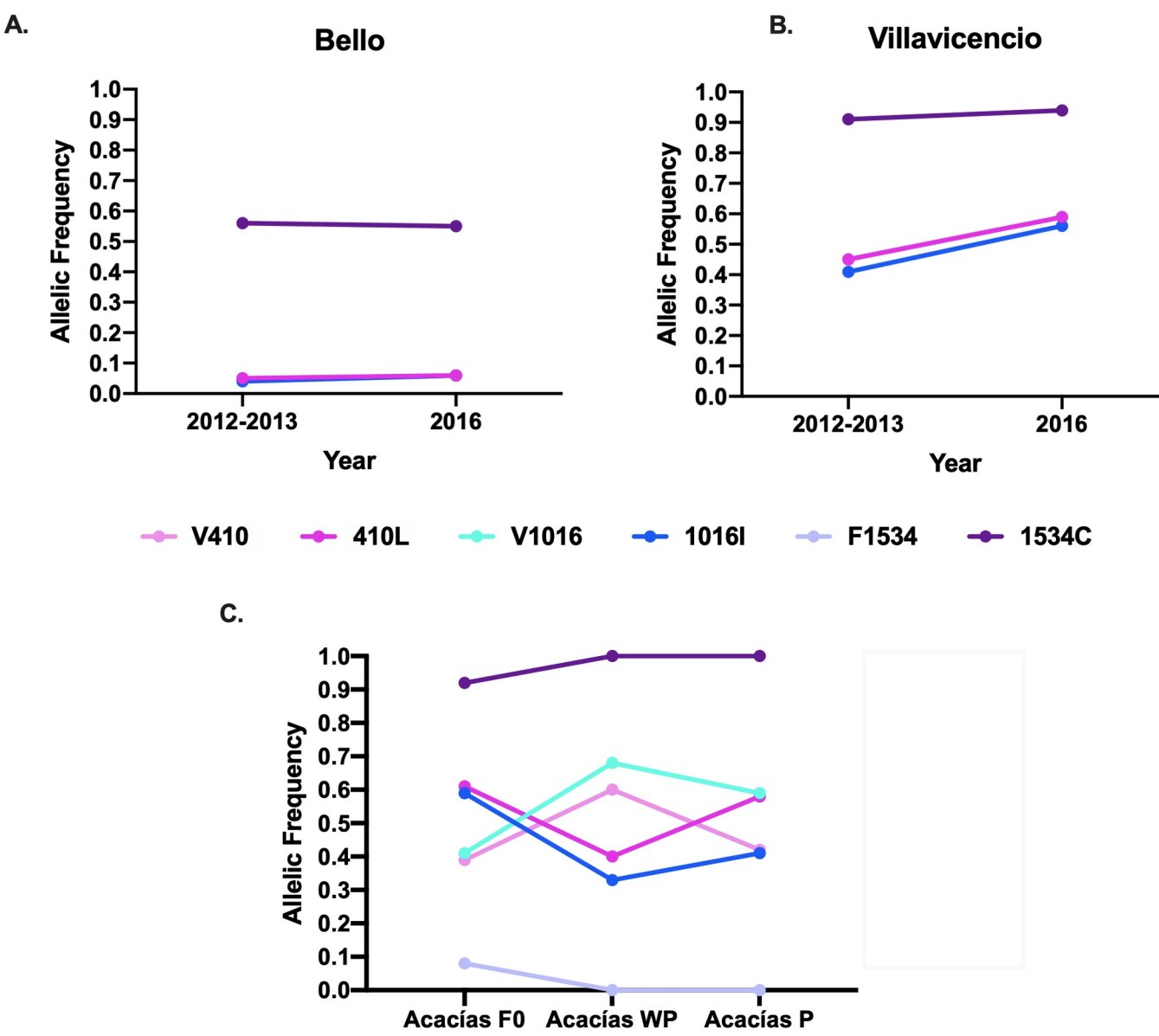

**Fig 3. Allelic frequencies for the positions V410L (pink), V1016I (blue), and F1534C (purple) in *Aedes aegypti* from three of the mosquito populations studied.** The light colors indicate the wt allele frequencies; the dark colors show the mutated allele frequencies. **A-B**. Time-course of allelic frequencies from Bello 2012–2013 (410 N = 58, 1016 N = 101, 1534 N = 57) and 2016 (410 N = 55, 1016 N = 59, 1534 N = 60) (A), and Villavicencio 2012 (410 N = 79, 1016 N = 89, 1534 N = 57 and 2016 (410 N = 50, 1016 N = 51, 1534 N = 51) (B) populations determined for mosquitoes collected in two different years. **C.** Allelic frequencies from Acacías population without lambda-cyhalothrin pressure (WP) and with lambda-cyhalothrin pressure (P) for six generations compared with the field-collected populations (Acacías F0).

410 and 1016 and heterozygous for loci 1534 ($VV_{410}/VV_{1016}/FC_{1534}$) or the triple heterozygous ($VL_{410}/VI_{1016}/FC_{1534}$). Likewise, only two individuals with the triple homozygous mutant genotype ($LL_{410}/II_{1016}/CC_{1534}$) were found in the selected population. Additionally, there was an increase in $VV_{410}/VV_{1016}/CC_{1534}$, especially in the population without selection pressure, and an increase in the combination of $LL_{410}/VI_{1016}/CC_{1534}$, particularly in the selected one. Finally, the new genotype heterozygous for loci 410 and double mutant homozygous for loci 1016 and 1534 ($VL_{410}/II_{1016}/CC_{1534}$) emerged in the population with heavy selection pressure with the insecticide.

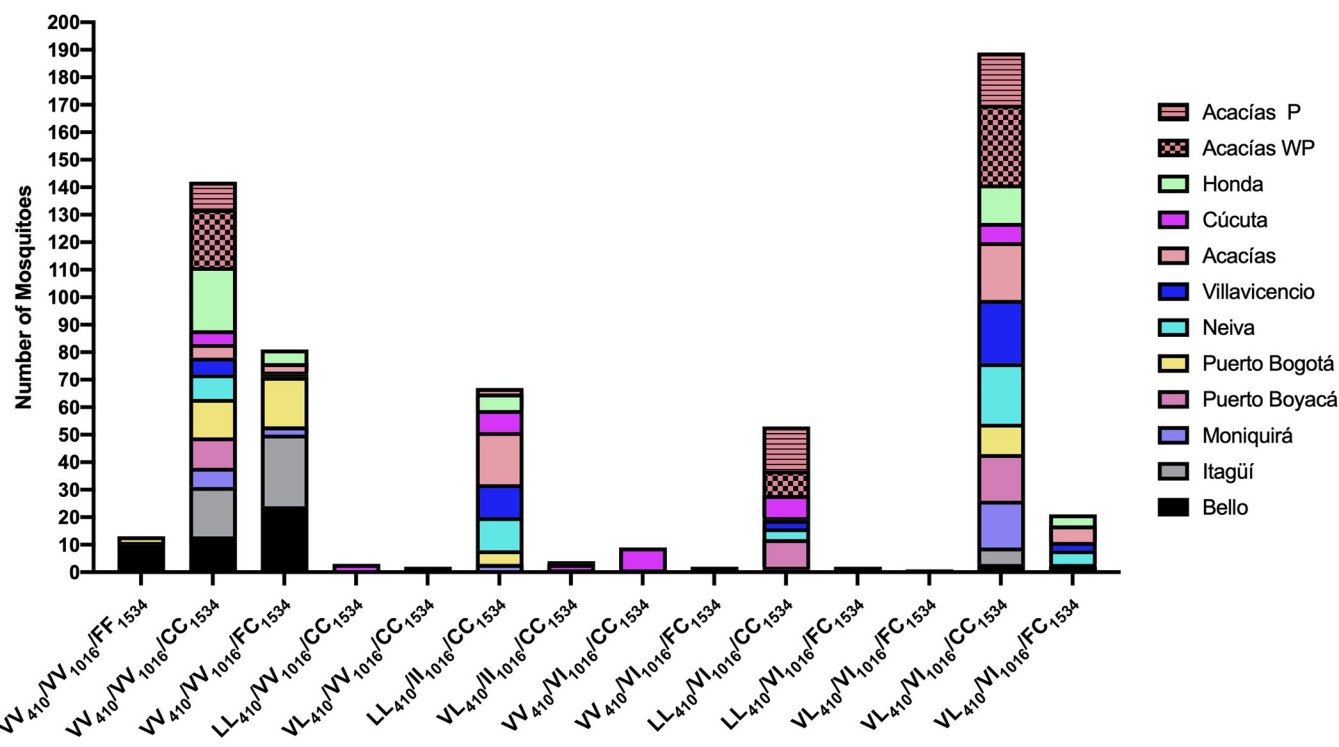

**Fig 4. Genotypes observed in Colombian *Aedes aegypti* populations.**

## 2.5. Association of different mutations and genotypes with pyrethroid resistance

A significant correlation was observed between the *kdr* alleles 410L, 1016I, and 1534C and resistance to lambda-cyhalothrin ($p = 0.0004$, 0.0036, and 0.0328, respectively). Remarkably, all the mosquito populations presented high frequencies of the 1534C *kdr* mutation, with the Puerto Boyacá and Cúcuta populations showing 100% of mosquitoes homozygous at this allele (Figs 2 and 5). Additionally, we found a significant positive correlation between the resistance ratio between lambda-cyhalothrin and the triple homozygous mutant genotype $LL_{410}/II_{1016}/CC_{1534}$ ($r = 0.739$; $p = 0.015$) and a negative correlation with other genotypes described in the supplementary S2 Table. Finally, we did not find a significant correlation between permethrin resistance ratio and any genotype (S2 Table).

## 2.6. Correlation between the different mutations

A robust linear relationship between the 410L and 1016I positions was found with $R^2 = 0.9445$ and a Pearson correlation of $p < 0.0001$. Additionally, a weaker but still significant linear relationship was observed between positions 410L and 1534C, and 1016I and 1534C ($R^2 = 0.7453$ and 0.6773, respectively; Spearman correlation values $p = 0.0065$ and 0.0328, respectively; Fig 6).

## 2.7. Biochemical assays

Based on the results obtained with *kdr* alleles, we wanted to analyze the activity of some insecticide-degrading enzymes between mosquitoes that underwent, or did not, selection pressure with the lambda-cyhalothrin insecticide. During six generations, the population exposed to

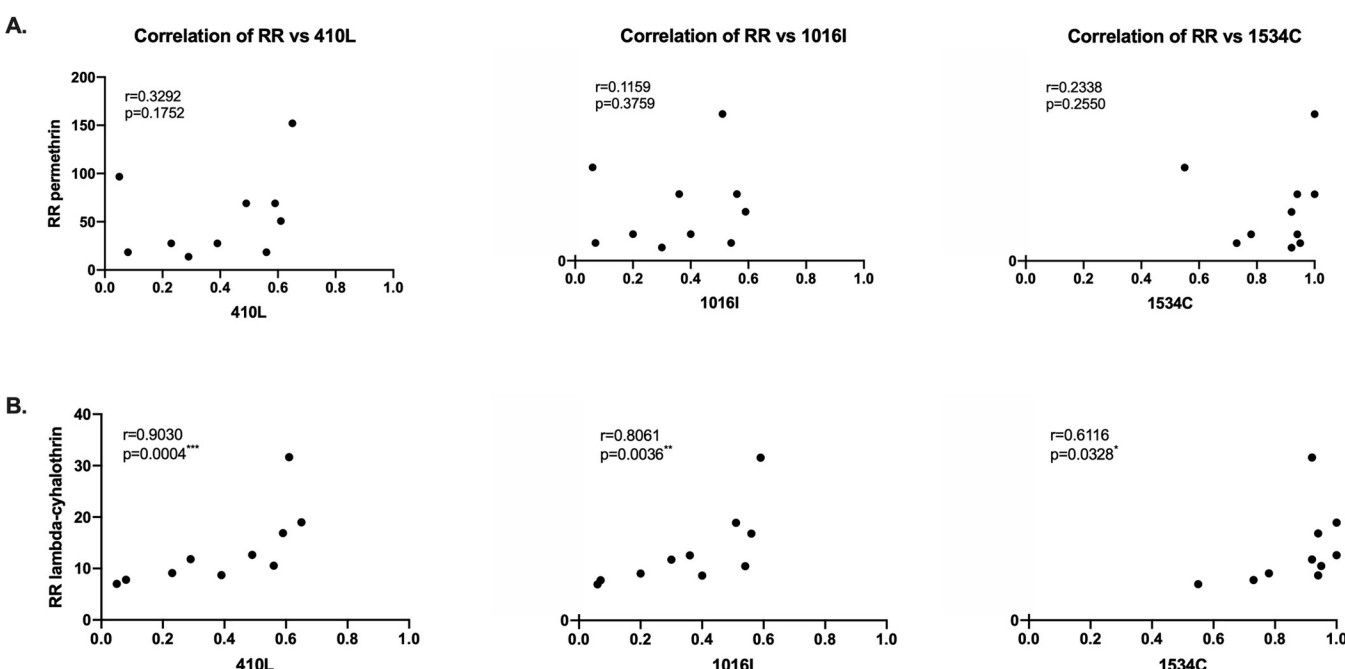

**Fig 5.** Spearman Correlation test (RR) for permethrin (A) and lambda-cyhalothrin (B) and the frequency of the mutated alleles for positions 410L, 1016I, and 1534C of the Colombian populations of *Aedes aegypti* evaluated. The Spearman correlation *r* and *p* values are shown for each correlation (*** = $p < 0.001$, ** = $p < 0.01$, * = $p < 0.05$).

insecticide showed significantly higher activity in the AChE and GST enzymes (Fig 7A and 7B). In contrast, higher β-EST activity levels were observed in mosquitoes from Acacías without selection pressure (Fig 7C). Both populations showed the same activity levels in the detoxification enzymes MFO and α-EST (Fig 7D and 7E). However, in multiple comparisons, all showed significant differences (S3 Table).

## 3. Discussion

The molecular mechanisms of insecticide resistance in *Ae. aegypti* are still not well understood. A clear understanding of the origin, evolution, and consequences of resistance to insecticides are needed to provide evidence-based solutions to the elevated risk of mosquito-borne diseases. Unfortunately, today there is no other alternative to insecticides, which means that insecticide-based mosquitoes control will still be used for a long time. Pyrethroids are the best

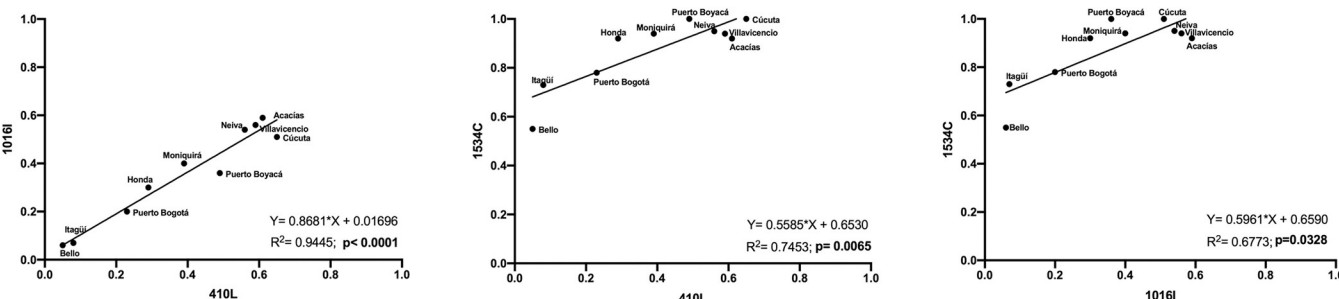

**Fig 6. Linear regressions of the sodium channel mutation frequencies at positions 410L vs. 1016I, 410L vs. 1534C, and 1016I vs. 1534C.** The data for 410L and 1016I passed normality tests, while the other comparisons did not pass (D'Agostino & Pearson and Shapiro-Wilk tests; alpha = 0.05).

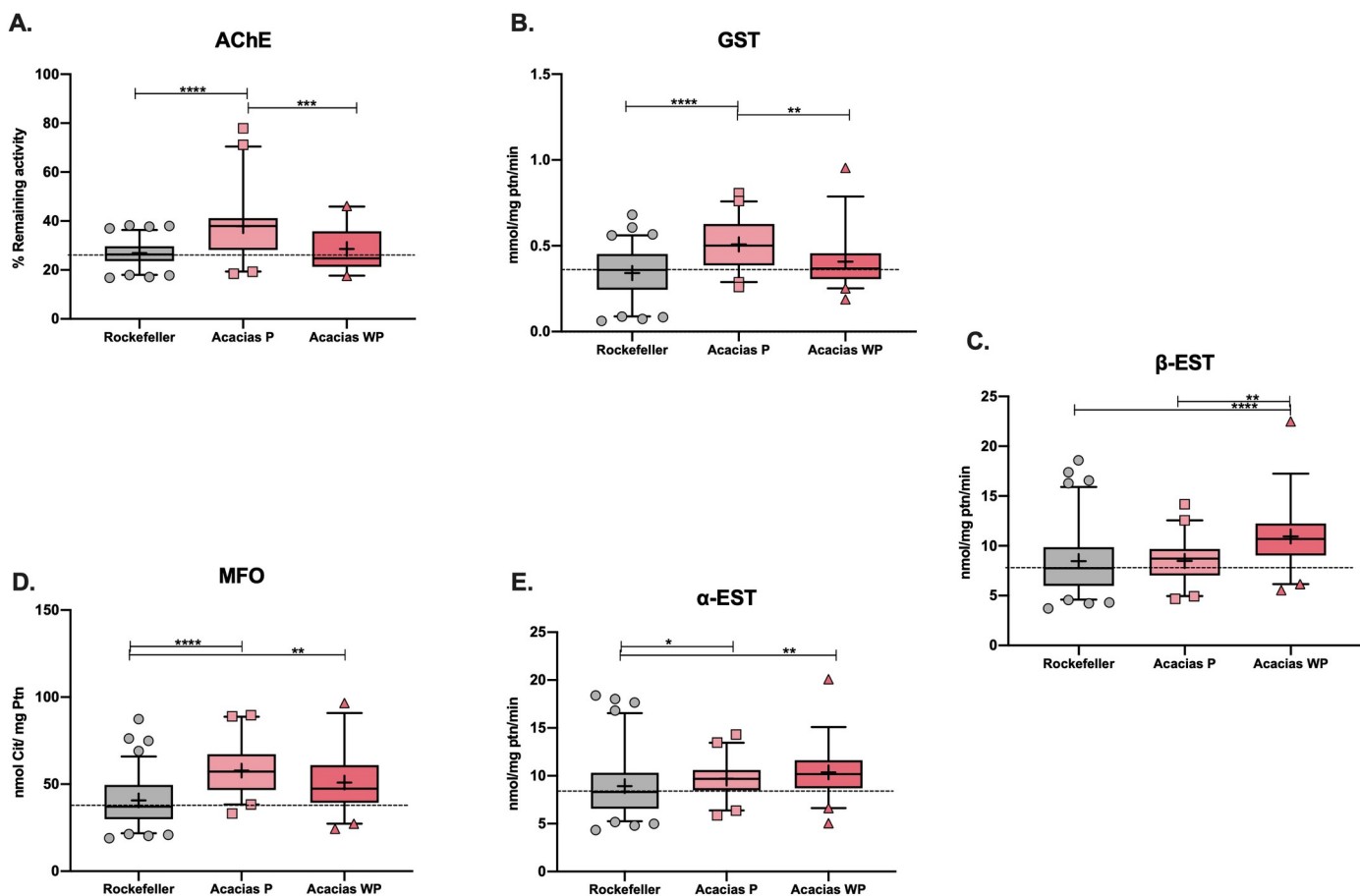

**Fig 7. Enzyme activity for *Aedes aegypti* collected from Acacías with lambda-cyhalothrin pressure in the lab (Acacías P) and without pressure (Acacías WP).**
A) Acetylcholinesterase (AChE), B) glutathione-S-transferases (GST), C) β-esterases (β-EST) D) mixed-function oxidases (MFO) and E) α-esterases, (α-EST). Forty individuals were used in each assay. Box plots include the mean (+), median (line), 5th, 25th, 75th, and 95th percentiles. The dotted line corresponds to the median activity for the susceptible strain (Rockefeller). Asterisks indicate significant differences between the Rockefeller strain and the corresponding population (Kruskal-Wallis test; **** = $p < 0.0001$, *** = $p < 0.001$, ** = $p < 0.01$, * = $p < 0.05$).

insecticide option because they are less toxic than carbamates and organophosphates to humans. In this study, we observed high pyrethroid (type I and II) resistance in ten Colombian *Ae. aegypti* populations, along with the presence of three different *kdr* mutations in the sodium channel. Moreover, we found an alteration in some insecticide-degrading metabolic enzymes in mosquitoes under insecticide selection pressure. In addition, we highlight changes in allele frequencies and the selection of some genotypes in insecticide-resistant mosquitoes.

At present, sixteen-point mutations have been reported in the voltage-gated sodium channel genes in *Ae. aegypti* from around the world [24–26]. For instance, in the southeast of China, the mutations S989P, V1016G, and F1534C have been reported [13], while in Yogyakarta, Indonesia, the mutations S996P, V1023G, and F1565C were found [27]. However, only three mutations, 410L, 1016I, and 1534C, in Colombia have been reported in *Ae. aegypti* populations across the country [14,18,19,28]. In this study, we confirmed the presence of these *kdr* alleles in Colombia and presented further evidence of changes in allelic frequencies in mosquitoes collected from ten different cities across seven departments of Colombia; most of the mosquitoes showed resistance to pyrethroid insecticides. This result is not surprising, as high resistance to different kinds of pyrethroids was recently reported in other regions in Colombia. All populations analyzed presented variations in these mutated alleles [19,20].

This paper presents further evidence indicating that the 1534C mutation is involved in the resistance to permethrin (type I-pyrethroid). All the highly resistant populations to permethrin also had a high allelic frequency of the mutation 1534C. Interestingly, we also found two mosquito populations where this allele was fixed (one-hundred percent of mosquitoes were homozygotes) and five others where the allele frequency was close to fixation with values above 92%. Similarly, high allelic frequencies of 1534C were also reported in mosquitoes from Venezuela, and it was found that the allele was fixed in some populations [29]. These results are not surprising since the correlation between the 1534C mutation and resistance to type I pyrethroids has been observed in other parts of the world [23,27,30,31].

In contrast, we propose that resistance to lambda-cyhalothrin requires the sequential evolution of at least three mutations. The high allelic frequency of mutant 1534C in all the studied Colombian populations indicates that this mutation appeared first, as a resistance response to type-I pyrethroids, probably due to cross-resistance to DDT used extensively to control mosquitoes that transmit malaria in Colombia [32–34]. Afterward, the 410L mutation surged, and finally, the 1016I mutation. This hypothesis is supported from the data obtained in mosquitoes from Cúcuta where $LL_{410}$ was observed together $VV_{1016}$ genotype. In this sense, our results support the idea proposed by Vera-Maloof et al., 2015 who indicated that the 1534C mutation appears first and that pyrethroid resistance requires the sequential evolution of new mutations [35]. Moreover, in other insect species, the combination of different mutations in the sodium channel gene produces the phenotype *super-kdr*, which confers higher resistance levels [36,37]. Whether these mutations mixed in Aedes confer high resistance levels in Colombian mosquito populations must be analyzed in more detail using crosses from congenic strains and examining changes in allele and genotype frequencies.

Synergism among point mutations in the sodium channel gene has been demonstrated before in *Ae. aegypti* [35]. Here, we observed that resistant mosquitoes are carrying two or three *kdr* mutations supporting this concept. For instance, we did not observe mosquitoes carrying only the 410L or 1016I mutations, but the mutation 1534C was found in all localities studied, where 29% (111/482) of mosquitoes presented only this mutation. Furthermore, the triple mutant homozygotes $LL_{410}/II_{1016}/CC_{1534}$ were found in 13.5% of the total number of the field caught mosquitoes, which consisted of individuals from all locations except Bello, Itagüí, and Puerto Boyacá and disappeared in the Acacías population without selection pressure and almost faded in the population pressured with the insecticide. In Malaysia, low frequencies of the triple homozygous mutations $GG_{1016}/CC_{1534}/PP_{989}$ (0.63%) have been reported, with researchers suggesting that the low occurrence of this triple homozygous mutation was most probably attributable to its effect on fitness [5,25]. This would suggest that the triple mutation $LL_{410}/II_{1016}/CC_{1534}$ may be linked to, or associated with, the loss of fitness parameters in *Ae. aegypti*. Finally, the triple homozygous wild-type ($VV_{410}/VV_{1016}/FF_{1534}$) was found only in low proportions in individuals from populations that were tolerant to lambda-cyhalothrin (Bello, Itagüí, and Puerto Bogotá).

To further explore the role of *kdr* mutations in pyrethroids resistance, we created a selection pressure with lambda-cyhalothrin on the Acacías mosquito populations over six generations. Interestingly, our data suggest that two mutations (410L and 1016I) are only meaningful in conferring low resistance levels since, after the $RR_{50}$ reaches values higher than 10, the allele frequencies do not change significantly (Fig 2). We support this idea with two findings: firstly, although the mosquitoes from Villavicencio, Neiva, and Acacías had the same frequencies of these *kdr* alleles, the mosquitoes from Acacías are more resistant to pyrethroids, indicating that the *kdr* genotype may not fully explain the variance in resistance phenotypes. Secondly, when the Acacías mosquito population was pressured with lambda-cyhalothrin for six generations, it became 450-fold resistant, but the frequencies of the 410L and 1016I mutant alleles in this population did not change significantly. These findings support the hypothesis that

additional resistance mechanisms are involved in the resistance phenotype. Nevertheless, the slight decrease in the allelic frequencies of these two mutations in the population without insecticide pressure and the change in the $RR_{50}$ from 32 in the parental population to 19 corroborates the importance of these mutations in the resistant phenotypes.

This current panorama in Colombian *Ae. aegypti* populations may result from decades of chemical control used in the vector surveillance programs without knowing the resistance status of the different populations. This phenomenon could favor resistant phenotypes in *Ae. aegypti* mediated by bottlenecks, as is the case for all populations evaluated for the insecticide permethrin (Table 1). Although not all populations were resistant to lambda-cyhalothrin, the presence of the *kdr* mutations and the frequent use of chemical control by health authorities could negatively select susceptible mosquitoes and the appearance of the resistant phenotype. The best example of this scenario is shown in the Bello mosquito population, which changed from being insecticide-susceptible to moderately insecticide-resistant in just three years.

Another good example is the Villavicencio population, which showed an increase in the frequencies of the mutated alleles in the sodium channel gene. Furthermore, in 2012, the Villavicencio population was in Hardy-Weinberg equilibrium [18], a condition that was lost in 2016, at least for the 1534 locus. This phenomenon is also supported by the analysis of inbreeding, in which we observed a deficit of heterozygotes.

Overall, our results suggest that other mutations in the sodium channel have not yet been identified or that other mechanisms are present in the Colombian mosquito populations. Recently, it was demonstrated that insecticide-resistant mosquitoes showed different expression profiles of genes involved in xenobiotic detoxification compared with non-resistant populations. These genes include glutathione-S-transferases, esterases, and cytochrome P450 mono-oxygenases [38]. This evidence highlights the importance of understanding both the genomic determinants of resistance (i.e., well-mapped mutations and copy number variations) and other mechanisms likely related to the crosstalk between metabolic activity and gene expression, which whole transcriptomic analyses can help to decipher [34]. In addition, the role of mosquito microbiota in insecticide resistance has begun to be studied. Recently, we analyzed the midgut microbiota from adult female *Ae. aegypti* collected in different Colombian locations to test for any correlations between insecticide resistance and specific microbial symbionts. Although the bacterial core was the same for all regions, interestingly, we detected differences in bacterial populations that might contribute to the insecticide-resistant phenotype [10]. According to our results, some species of bacteria such as Klebsiella sp and *Pseudomonas oleovorans* presented the ability to degrade lambda-cyhalothrin insecticide [39]. Similarly, the genus Rhizobium was related to the degradation of the malathion insecticide [40]. Moreover, pyrethroids degradation based on bacteria such as Acinetobacter, Bacillus, Ochrobactrum, Pseudomonas, Serratia, Sphingomonas, and Klebsiella has been recently reported [41].

Since the insensitivity of *Ae. aegypti* to pyrethroids is also associated with higher activity and expression of detoxifying enzymes; we also evaluated some of these enzymes in the Acacías populations with or without insecticide selection pressure. We found that esterases and mixed-function oxidases were related to pyrethroid resistance, as was previously reported in Colombian populations from Guadalupe (Huila department), Tumaco (Nariño department), Valledupar (Cesar department), Montería (Córdoba department), Juan de Acosta (Atlántico department), Medellín (Antioquia department) and Yumbo (Valle del Cauca department) [19,42]. Furthermore, mosquito populations around the world are showing similar behaviors [6,29,43–45]. In addition, we found that Glutathione S-Transferase activity is increased in the population exposed to the insecticide, supporting its suggested role in resistance. Recently, Aponte et al. (2020) reported that metabolic resistance and *kdr* mutations were also present in pyrethroid resistant mosquitoes, but they found enhanced GST expression [19].

On the other hand, although acetylcholinesterase (AChE) altered activity is not associated with pyrethroids resistance, we found that our selected population, through exposure to insecticide, presented altered levels of this enzyme. This result has also been observed in bees, where exposure to deltamethrin increased AChE activity, suggesting that AChE activity could function as a biomarker of insecticide exposure [46]. The alteration of enzyme activity could result from mutations in other genes from permanent exposure to insecticides. These questions merit further study using other tools such as genome and RNA-seq analyses.

Finally, we hypothesized that the underlying mechanism of insecticide resistance is multifactorial, involving genetic, biochemical, and possibly epigenetic aspects. Future studies involving analyses of transcriptomes, genomes, microbiomes, and epigenomes from field resistant and susceptible *Ae. aegypti* are needed to understand this phenotype fully.

In conclusion, our results indicate that the continuous exposure of *Ae. aegypti* to insecticides could favor the fixation of some *kdr* alleles and the emergence of some genotypes involved in insecticide resistance, as was observed in the Acacías mosquito populations. In addition, the results observed in the Villavicencio population demonstrate that in just four years, the *kdr* allele frequency was altered, and the mosquitoes became more resistant to insecticides. The fixation of the 1534C allele and its connection with resistance to permethrin suggest that this insecticide should be monitored continuously. These Colombian mosquito populations offer the potential to study variations in insecticide resistance and shed light on the origins, evolution, mechanisms, and management of insecticide resistance. These data indicate that health authorities undertake permanent molecular surveillance programs to identify these mutations and carry out frequent bioassays in mosquitoes from the field to select which insecticides should be used to control and prevent mosquito-borne diseases. Finally, further studies are necessary to monitor the changes in allelic frequencies of these mutations, identify other mechanisms involved in insecticide resistance, and improve disease control and prevention programs.

## 4. Materials and methods

### Ethics statement

Ethical approval (Act No 113 of 2017) for analyzing animal species was obtained from the Antioquia University's animal ethics committee. Moreover, the University of Antioquia has a permit from the national environmental authority to collect biological specimens for research purposes (0524 -27-05-2014).

### 4.1. Study area

This study was conducted during 2016–2019 in ten Colombian municipalities from seven departments. We focused on cities that had experienced increased dengue transmission or outbreaks in the last few years. We chose 20 randomized houses from four to six neighborhoods in each municipality following the recommendations previously published [47]. (Table 2 and Fig 1).

### 4.2. Ae. aegypti collections

*Aedes* spp. larvae, pupae, adults, and eggs were collected in collaboration with staff involved in vector-borne diseases programs for each city. The immature stages were reared to adults (F0 generation) and maintained under controlled conditions of temperature (28°C ± 1°C), relative humidity (80% ± 5%), and photoperiod (12 h light: 12 dark). After adult identification, mosquitoes were separated by species and *Ae. aegypti* specimens were kept for breeding following

**Table 2. Eco-epidemiological information for *Aedes aegypti* populations collected from seven provinces.**

| Province | City | Altitude | Annual Biotemperature average | Annual rainfall | Year of collection | Number of dengue cases during collection by province | GPS coordinates | |
|---|---|---|---|---|---|---|---|---|
| Antioquia | Bello (1) | 1310 m.a.s.l | 26.7˚C | 1347 mm | 2016 | 1739 | 6˚19′55″ N | 75˚33′29″ O |
| | Itagüí (2) | 1550 m.a.s.l. | 21.1˚C | 1760 mm | 2018 | 3956 | 6˚11′4.6" N | 75˚ 35'56.9" O |
| Boyacá | Moniquirá (3) | 1669 m.a.s.l. | 19.6˚C | 2201 mm | 2017 | 378 | 5˚52'35" N | 73˚ 34'22.2" O |
| | Puerto Boyacá (4) | 130 m.a.s.l. | 27.7˚C | 2369 mm | 2019 | 944 | 5˚58'32.4" N | 74˚ 35'32.1" O |
| Cundinamarca | Puerto Bogotá (5) | 240 m.a.s.l. | 27.5˚C | 1598 mm | 2016 | 4546 | 5˚12'22.8" N | 74˚ 43'51.5" O |
| Huila | Neiva (6) | 442 m.a.s.l. | 27.1˚C | 1216 mm | 2016 | 4163 | 2˚55'38.3" N | 75˚ 16'54.8" O |
| Meta | Villavicencio (7) | 467 m.a.s.l. | 25.5˚C | 3856 mm | 2016 | 2607 | 4˚8'31.2" N | 73˚ 37'35.9" O |
| | Acacías (8) | 498 m.a.s.l. | 25.0˚C | 3247 mm | 2016 | 2607 | 3˚59'13" N | 73˚ 45'28.7" O |
| Norte de Santander | Cúcuta (9) | 320 m.a.s.l. | 26.6˚C | 622 mm | 2017–2018 | 1240–4874 | 7˚53'38.1" N | 72˚ 30'28.2" O |
| Tolima | Honda (10) | 226 m.a.s.l. | 27.5˚C | 1608 mm | 2016 | 5853 | 5˚12'30.8" N | 74˚44'9" O |
| m.a.s.l.: meters above sea level | | | | | | | | |

standardized techniques [48]. F1 larvae were used for the bioassays described below with the lambda-cyhalothrin insecticide, and larvae up to F6 were utilized with the permethrin insecticide. The insecticide susceptible Rockefeller strain was used as a control in all the experiments.

## 4.3. Bioassays

Larval bioassays were carried out according to WHO guidelines [49] using technical grade permethrin (96.1% active ingredient [a.i.]) and lambda-cyhalothrin (99.8% a.i.), purchased from Sigma-Aldrich (USA). Although pyrethroids are not used for larval treatment, we tested them against *Ae. aegypti* larvae to obtain information on the larval resistance status that may reflect the adult resistance status since pyrethroids' target is a constitutive gene. Bioassays were performed on late third- and early fourth-instar larvae of each population in containers with 99 ml of distilled water and one ml of the insecticide tested at the desired concentration. Three replicates per concentration (20 larvae per replicate) and six concentrations in the activity range of each insecticide were used to determine the lethal concentrations ($LC_{50}$ and $LC_{90}$), and confidence intervals were calculated for every mosquito population included in the three biologically independent replicates [20,50]. We used lambda-cyhalothrin concentrations ranging from 0.0009 to 0.06 and permethrin concentrations from 0.0007 until 0.096. The susceptible Rockefeller laboratory *Ae. aegypti* strain was used as a reference population. Control treatments consisted of 1% ethanol, and larval mortality was recorded after an exposure of 24 hours. For each bioassay, temperature, relative humidity, and photoperiod were maintained as described above.

## 4.4. Changes in allelic frequencies in mosquitoes with or without insecticide pressure

Since the Acacías population showed the highest resistance to lambda-cyhalothrin, it was selected to assess allele frequency changes with or without the insecticide selection pressure.

For the first experiment, late third- and early fourth-instar larvae from the F1 generation were exposed to the $LC_{50}$ (0.015ppm), and the surviving larvae were reared to adults. Larvae of the next generation (F2) underwent the same experimental procedures. Subsequently, the same experiment was performed in larvae from F3- F6 generations, but these were exposed $LC_{90}$ (0.05) until the F7 generation was reached. Next, the RR to lambda-cyhalothrin and permethrin was determined using the larval bioassays described above. For the second experiment, mosquitoes from the F1 generation of the Acacías population were maintained without selection pressure with the insecticide until generation F7 was obtained, when the larval bioassay was performed. The resistance level and *kdr* mutations were quantified in all cases.

### 4.5. Kdr mutations genotyping

Allele-specific PCR (AS-PCR) was used to identify the V410L, V1016I, and F1534C *kdr* mutations because they were found previously in insecticide-resistant mosquitoes from Colombia [18,20]. Genomic DNA from individual mosquitoes was extracted using the protocol described by Collins et al. [51]. A minimum of 30 adult mosquitoes from each population was analyzed. Likewise, mosquitoes from the Acacías population with and without insecticide selection pressure were processed for genotyping using AS-PCR as described previously [18]. The susceptible *Ae. aegypti* Rockefeller strain was used as a reference to the wild-type alleles (V1016, V410, and F1534) of the voltage-gated sodium channel gene.

### 4.6. Biochemical assays

To determine if the observed insecticide resistance was also due to metabolic resistance, the activity levels of enzymes associated with resistance were measured. Enzymatic activity assays were conducted on Acacías *Ae. aegypti* populations described above with or without the insecticide lambda-cyhalothrin pressure. Biochemical assays were performed following the guidelines reported previously with slight modifications [52]. Briefly, two to three-day-old females (40 in total) were homogenized individually in 300 μL of deionized water on ice. The adult mosquitoes were assayed for acetylcholinesterase (AChE), mixed-function oxidases (MFO), α-esterases (α-EST), β- esterases (β-EST), and glutathione-S-transferases (GST) activities. For this process, 25 μL of the homogenate was pipetted for the AChE assay and 20 μL for the MFO assay. Subsequently, samples were centrifuged at 14,000 rpm for 60 seconds at 4˚C; the supernatant was aliquoted and transferred to 96-well microplates for all other enzyme assays. Absorbances were measured using an ELISA Multiskan Spectrum from Thermo Fisher Scientific, using the wavelengths reported previously for each enzyme [6]. Five individuals from the susceptible Rockefeller strain were included in every experiment as controls.

Finally, since the body masses between mosquitoes are different, all analyses of enzyme activities were corrected using the total protein concentration as a standard correction factor. The commercial protein assay kit (Pierce BCA Protein Assay Kit, Thermo Scientific, Rockford, IL) was used with 10 μL of mosquito homogenate following manufacture recommendations.

### 4.7. Data analysis

**4.7.1 Bioassays.** For each population and each insecticide, the dose and mortality ratios were adjusted using a regression (p< 0.05). Results were analyzed using the log-probit analysis to estimate the regression line's slope and determine the 50% and 90% lethal concentrations ($LC_{50}$ and $LC_{90}$, respectively) with 95% confidence intervals (CIs). The resistance ratios ($RR_{50}$ and $RR_{90}$) were obtained by comparing the $LC_{50}$ or the $LC_{90}$ of the field populations and Rockefeller strain. Field populations were considered moderately resistant to a given

insecticide when their Resistance Ratios (RR) were between 5 and 10 and very resistant when the RR values were over 10 [49].

**4.7.2. Allelic and genotypic frequencies of kdr mutations.** GENEPOP v.4.6 (Laboratoire de Genetique et Environment, Montpellier, France, http://genepop.curtin.edu.au/) was used to calculate allelic and genotypic frequencies, and the data were tested for conformity to the Hardy–Weinberg equilibrium [53]. The Chi-square test was used to determine whether the populations were in Hardy-Weinberg equilibrium. If the calculated value of $\chi^2$ was < tabulated $\chi^2$ (1 gl) = 3.84 and p> 0.05, the $H_0$ was accepted, meaning that the study population was in HW equilibrium; otherwise, the $H_a$ was accepted. Moreover, the endogamy coefficient, $F_{IS}$, was calculated following the method previously reported by Pareja-Loaiza et al. 2020 [20]. Finally, the number of mosquitoes with every haplotype for the three *kdr* mutations was calculated for every population.

**4.7.3. Association of kdr alleles with pyrethroid resistance and between alleles.** The association between *kdr* allele frequencies or genotypes and pyrethroid susceptibility profiles was tested using a Spearman correlation test (p <0.05). Additionally, the relationship between two different alleles was evaluated using a simple linear regression and either a Pearson or Spearman correlation test (p <0.05) according to the normality of the data. Finally, a Bonferroni test correction was performed for multiple comparisons. Both analyses were performed in the GraphPad Prism program (version 5.1 for Windows, GraphPad Software, La Jolla California USA, www.graphpad.com) and SPSS for the Bonferroni correction.

**4.7.4. Biochemical assays.** The biochemical assay was analyzed using Kolmogorov-Smirnov tests to check the normality of the enzymatic activities. Differences in enzymatic activities between each field population and the Rockefeller strain were compared using the Mann-Whitney non-parametric test. The comparison between groups was performed using the Kruskal-Wallis test in the GraphPad Prism program (version 5.1 for Windows, GraphPad Software, La Jolla California USA, www.graphpad.com).

## Supporting information

**S1 Table. Genotypic and allelic frequencies of the V410L, V1016I, and F1534C *kdr* alleles in Colombian mosquito populations.** The HW equilibrium and the coefficient of endogamy ($F_{IS}$) are shown.
(DOCX)

**S2 Table. Genotypes observed in Colombian *Aedes aegypti* populations and the association between genotypes frequencies and pyrethroid susceptibility profiles using the Spearman correlation test.**
(DOCX)

**S3 Table. Mean of enzyme activities for Acetylcholinesterase (AChE), glutathione-S-transferases (GST), β-esterases (β-EST), Mixed function oxidases (MFO), and α-esterases (α-EST), detected in *Aedes aegypti* from Acacías with and without insecticide pressure.** P-values were calculated using the Kruskal-Wallis test for multiple comparisons.
(DOCX)

## Acknowledgments

The authors want to thank Prof. Carl Lowenberger from Simon Fraser University and Jenny Peterson from Portland State University for the English edition.

## Author Contributions

**Conceptualization:** Ana María Mejía-Jaramillo, Omar Triana-Chávez.

**Data curation:** Yurany Granada, Ana María Mejía-Jaramillo, Omar Triana-Chávez.

**Formal analysis:** Yurany Granada, Ana María Mejía-Jaramillo, Sara Zuluaga, Omar Triana-Chávez.

**Funding acquisition:** Omar Triana-Chávez.

**Investigation:** Yurany Granada, Ana María Mejía-Jaramillo, Sara Zuluaga.

**Methodology:** Yurany Granada, Ana María Mejía-Jaramillo, Sara Zuluaga, Omar Triana-Chávez.

**Supervision:** Omar Triana-Chávez.

**Writing – original draft:** Yurany Granada, Ana María Mejía-Jaramillo, Sara Zuluaga, Omar Triana-Chávez.

**Writing – review & editing:** Ana María Mejía-Jaramillo, Omar Triana-Chávez.

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
