## [Decision Letter · Decision Letter 0]

12 Oct 2021

Dear Dr Triana-Chávez,

Thank you very much for submitting your manuscript "Molecular Surveillance of Resistance to Pyrethroids Insecticides in Colombian Aedes aegypti Populations" for consideration at PLOS Neglected Tropical Diseases. As with all papers reviewed by the journal, your manuscript was reviewed by members of the editorial board and by several independent reviewers. The reviewers appreciated the attention to an important topic. Based on the reviews, we are likely to accept this manuscript for publication, providing that you modify the manuscript according to the review recommendations. 

All reviewers agreed that this study is important and of good quality. Please respond to the suggestions for improvement point by point when a revised version is submitted.

Sincerely,

Jeffrey H Withey

Associate Editor

Tereza Magalhaes

Deputy Editor

Your manuscript has been reviewed by three experts in the field and all agree that this study is important and of good quality. Please respond to the suggestions for improvement point by point when a revised version is submitted.

Reviewer's Responses to Questions

**Key Review Criteria Required for Acceptance?**

**Methods**

-Are the objectives of the study clearly articulated with a clear testable hypothesis stated?

-Is the study design appropriate to address the stated objectives?

-Is the population clearly described and appropriate for the hypothesis being tested?

-Is the sample size sufficient to ensure adequate power to address the hypothesis being tested?

-Were correct statistical analysis used to support conclusions?

-Are there concerns about ethical or regulatory requirements being met?

Reviewer #1: 1. Are the objetives of the study claerly articulated with a clear testable hypothesis stated?

The objectives of the study are articulated with the hypothesis of the study, however, it is suggested to expand them taking into account the results. Include the above at the end of the introduction section.

2 Is the study design appropriate to address the stated objectives?

The design of the study is perfectly in line with the results obtained, however, it is suggested to expand the objectives of the study (lines 78 to 85) taking into account the design and the results obtained.

3. Is the population clearly described and appropriate for the hypothesis being tested?

The study population is clearly described and appropriate for the hypothesis raised.

4.Is the sample size sufficient to ensure adequate power to address the hypothesis being tested?

The size of the sample used for the development of each of the bioassays is adequate, as well as each of the statistical tests used.

5. Were correct statistical analysis used to support conclusions?

The statistical analysis is correct to support the results and the conclusions obtained.

6.Are there concerns about ethical or regulatory requiremenst being met?

In the Materials and Methods section, numeral 4.8, the ethical requirements are clarified "Statement of Ethics Ethical approval was obtained (Law No. 113 of 2017) for the analysis of animal species from the animal ethics committee of the University of Antioquia"

-Please, in the Materials and Methods section, provide information about the permits you obtained for the development of the work. Include the full name of the authority that approved access to the field site. If permits are not required, explain why briefly.

-Was the selection of the study site justified based on previous results of resistance monitoring at the national level? This should be mentioned somewhere in the manuscript.

-Please make sure to use proper abbreviation for species name throughout the manuscript (e.g. Aedes aegypti being Ae. aegypti instead of A. aegypti).

-In the Materials and Methods section 4.2 lines 409 and 410 please include the name of the protocol or the appointment in which the technique is evidenced.

-In the Materials and Methods section 4.3, document through other studies the idea consigned in lines 417 to 419.

-In the Materials and Methods section 4.3, include the six concentrations evaluated lines 422,423.

-In the Materials and Methods section section 4.4, the insecticide permethrin was also used with the Acacias population for showing a high degree of resistance as indicated in lines 431, 432 and 433 for lambda-cyhalothrin? Please clarify.

Reviewer #2: Lines 487-489: “If the calculated value of chi-squared was < tabulated chi-squared (1 gl) = 3.84 and p< 0.05, the H0 was accepted, meaning that the study population was in HW equilibrium; otherwise, the Ha was accepted.” The inequality sign for the p-value is flipped in the text, and should be edited to “p> 0.05” for the statement to be correct. 

Lines 206-209 and Supplementary Table S2: Given the large number of Spearman correlation analyses performed (14 triple genotypes vs each insecticide, 28 analyses total), the p-values should be corrected for multiple comparisons. For instance, the sole correlation between triple genotype and permethrin resistance (VV410/VV1016/CC1534) may no longer meet the threshold for significance after multiple comparison adjustment. In contrast, the identified significant correlations for lambda-cyhalothrin look more robust (with the possible exception of VV410/VI1016/FC1534).

Reviewer #3: The study objectives were clear, appropriately designed with a clearly defined population and sufficient sample to test the stated objectives. The statistical analyses were appropriate and in line with field standards. There do not appear to be concerns on ethical or regulatory grounds.

**Results**

-Does the analysis presented match the analysis plan?

-Are the results clearly and completely presented?

-Are the figures (Tables, Images) of sufficient quality for clarity?

Reviewer #1: 1. Does the analysis presented match the analysis plan?

The analysis presented is in accordance with the one proposed in the Materials and Methods section, numeral 4.7

2. Are the results clearly and completely presented?

The results were presented clearly and completely.

3. Are the figures (Tables, Images) of sufficient quality for clarity?

Figure 1 is not seen defined, please consider improving resolution since the number inside the semicircle is not clearly displayed. Consider deleting the image in the upper left of the figure and placing the map of South America on the left.

Figures 4 and 6 are not defined, please consider improving resolution.

Supplementary Table 1 Specify in the Table’s footnotes what ND means in the FIS column

In figure 3 Include the N used to determine the allele frequencies of the populations of Bello and Villavicencio during 2012 to 2016, lines 151 to 153

- In the Results section, numeral 2.1 on lines 93 and 94 please consider clarifying that the data correspond to permethrin. Additionally, consider including the itagui data with RR50 of 18.43, lines 93 and 94.

- In the Results section, numeral 2.2, please clarify if the results mentioned in lines 137 to 147 obtained in the study carried out in Bello and Villavicencio during the years 2012 to 2016 all correspond to the same study "A Point Mutation V419L in the Sodium Channel Gene from Natural Populations of Aedes aegypti Is Involved in Resistance to lambda-Cyhalothrin in Colombia. Insects. 2018; 9 (1), if so, place the quote (18) at the end of the paragraph.

- It is important to clarify in the Discussion section why the V410L mutation was named as V419L in the article “A Point Mutation V419L in the Sodium Channel Gene from Natural Populations of Aedes aegypti Is Involved in Resistance to lambda-Cyhalothrin in Colombia”.

- In the Discussion section, expand Bello's situation regarding the use of insecticides three years ago and currently, lines 324 to 329. Additionally, include data that support the idea that in 2012 the population of Villavicencio was on HW equilibrium, lines 330 to 332.

- In the Discussion section Include more studies in which the microbiota of the A. aegypti midgut is correlated with resistance to insecticides, lines 344 to 349

Reviewer #2: Lines 121-122 and Supplementary Table S1: For F1534C, Itagui does not meet the criteria to reject the null hypothesis. The chi-squared value is < 3.84 and p = 0.050 (but is not less than 0.05). Instead, the authors can revise the text to state that the value for Itagui “approached significance”. 

Lines 121-122: The authors should not comment on Puerto Boyaca or Cucuta Hardy-Weinberg equilibrium (HWE) because HWE analysis does not apply for alleles that have achieved fixation. The authors can revise the text to simply state that for those two locations, the L allele achieved fixation (and not mention anything about HWE). 

Lines 123-130: The authors should only comment on inbreeding coefficients for populations that are in HW disequilibrium, because these are the only populations where the genotype frequencies are significantly different from what we would expect at HWE. For populations that have achieved HWE, to state that there is a heterozygote “deficiency” or “excess” is not meaningful. HWE (acceptance of the null hypothesis) implies that the genotype frequency is not significantly different from the equilibrium state, so even if there is a higher or lower number of heterozygotes than expected, the difference is not significant.

Figure 6: Please add the p-value of each correlation to the figure, for ease of reading.

Reviewer #3: The analyses described matched the proposed analyses and the results were presented quite well in a variety of graphs and tables with sufficient clarity to appreciate their findings.

**Conclusions**

-Are the conclusions supported by the data presented?

-Are the limitations of analysis clearly described?

-Do the authors discuss how these data can be helpful to advance our understanding of the topic under study?

-Is public health relevance addressed?

Reviewer #1: 1. Are the conclusions supported by the data presented?

The conclusions are clearly supported by the data presented.

2. Are the limitations of analysis clearly described?

Please include in the manuscript at the end of the Discussion section the strength and limitations of the study

3. Do the authors discuss how these data can be helpful to advance our understanding of the topic under study?

Include in the manuscript in the Discussion section how the variations in the Kdr frequencies affect the control methods currently used in the study site.

4. Is public health relevance addressed?

Please include in the manuscript at the end of the Discussion section the relevance of this type of studies for public health.

Reviewer #2: Lines 272-275 and Supplementary Table S1: In reference to 1534C, “Interestingly, we also found two mosquito populations where this allele was fixed (one-hundred percent of mosquitoes were homozygotes) and two others where the allele frequency was close to fixation with values above 95%.” Reviewing the table, I see only one population (Neiva) where the allele frequency is 0.95 or higher but has not achieved fixation, the other populations are at 0.94 and below. Please correct this text in the manuscript. 

Lines 284-287: The authors propose that after the emergence of the 1534C mutation, that 410L emerged next, followed by 1016I, however it is not clear why the authors’ data supports the emergence of 410L then 1016I, in that temporal order. While I agree 1534C is most likely the first mutation, it is plausible that 1016I could emerge first, followed by 410L. Alternatively, 410L and 1016I could co-evolve in the same population contemporaneously. Please revise this section to explicitly demonstrate how your data support a specific temporal order of mutation emergence, or state that multiple models for the order of mutation emergence are consistent with your results.

Lines 290-292: The finding that the authors never observed solo mutations in either 410L or 1016I is one of the most notable results of the study, while “1534C was found in all localities studied.” In the Discussion, please expand on your interpretation of why this pattern of mutations emerged. Do you feel that 1534C was most prevalent simply because of prior DDT exposure and cross-resistance (lines 283-284)? Or is there an epistatic interaction between 1534C and the other two mutated loci? For example, does the presence of 1534C somehow (a) increase the probability of the 410L or 1016I mutations occurring, or (b) convey a fitness advantage that allows these mutations to persist, whereas in isolation the fitness cost of 410L or 1016I is too high to maintain these mutants in the population? The Discussion would be strengthened by hearing the authors’ thoughts on these possible mechanisms.

Reviewer #3: The conclusions were supported by the data and the limitations were noted. The authors described the public health relevance of their findings and how they were advance the field.

**Editorial and Data Presentation Modifications?**

Reviewer #1: Accept

Reviewer #2: (No Response)

Reviewer #3: Accept with minor revisions. I have only a few specific edits/critiques:

-The logical flow of the paper might be enhanced by presenting the 'Material and Methods' section of the paper ahead of the 'Results' section, as understanding of the results necessitated being familiar with what was accomplished first. 

-For Line 270, I would alter the word 'new' to read 'additional' or something along those lines, as the sentence last sentence in the paragraph seems to contradict the first sentence in the paragraph. 

-For Line 410, please describe or include a reference for the 'standardized techniques' of breeding that are alluded to.

-For Table 1, please include a footnote for what 'm.a.s.l.' stands for.

-Please correct 'mosquitos' to the correct English plural 'mosquitoes' throughout the paper (e.g. Lines 8, 129, 257, 430).

-Please correct 'CL50' or 'CL 90' to 'LC50' or 'LC90', (e.g. Lines 160, 434, 437).

-Please correct 'correlation test of Spearman' to the active voice 'Spearman correlation test' (e.g. 213, 495, 497, Supplementary Table S2).

Additional minor edits are included in the attached document.

**Summary and General Comments**

Reviewer #1: The present study is of great importance since it expands the knowledge of the susceptibility status of A. aegypti to pyrethroid-type insecticides used for control in Colombia, as well as the knowledge of the different resistance mechanisms and their modes of action

Reviewer #2: This manuscript by Garzon et al. describes a field survey of pyrethroid resistance in Aedes aegypti populations from ten different geographic locations in Columbia that have a high incidence of dengue. The authors expand on their field data using genotypic and biochemical assays to explore potential mechanisms for the resistance phenotypes they observe. 

My critiques and suggestions are limited to minor corrections, modifications, or clarifications. Overall, this is a strong manuscript, due to the scope of the project, the use of a range of complementary experimental methods (phenotypic bioassays, allele-specific PCR, and enzymatic activity assays), and several notable discoveries, including (1) the high prevalence and geographic variability of pyrethroid resistance in Columbia, (2) evidence of selection for specific combinations of kdr mutations, (3) the correlation of kdr mutant alleles with lambda-cyhalothrin resistance and with the presence of other kdr mutations, (4) that in field populations the 410L and 1016I do not occur in isolation, but only in combination with mutations at other kdr loci, (5) allele frequency alone is insufficient to explain the high levels of pyrethroid resistance, and (6) increased metabolic enzyme activity that may account for part of the resistance mechanism. I also applaud the authors for the clarity of their visual data presentation, particularly Figures 1 through 4, which condensed a lot of numerical data into visually clean and comprehensible images. 

I recommend Minor Revisions before the manuscript is accepted for publication.

Reviewer #3: Overall, I found the paper to be an excellent addition to the current knowledge of pyrethroid resistance. The focus on the genetic and toxicological makeup of the populations found throughout Colombia will be a boon to local efforts as well as to the larger field dealing with similar issues of control. I especially appreciated the time course observations of the mosquito populations samples over several years, and the loss of function/gain of function analyses employed with rearing out to 7 generations both with and without insecticidal pressure. The discussion appeared to be well-versed in the current literature, so my only critique in this area would be that the authors did not reference the phenomenon of super-kdr (either in favor or in opposition to the concept) when discussing synergistic phenotypes.

PLOS authors have the option to publish the peer review history of their article (what does this mean?). If published, this will include your full peer review and any attached files.

Reviewer #1: Yes: Paula Pareja-Loaiza

Reviewer #2: Yes: Joshua R. Lacsina

Reviewer #3: Yes: Natasha Marie Agramonte, PhD

Figure Files:

Data Requirements:

Reproducibility:

References

---

## [Editor Report · Decision Letter 1]

16 Nov 2021

Dear Dr Triana-Chávez,

We are pleased to inform you that your manuscript 'Molecular Surveillance of Resistance to Pyrethroids Insecticides in Colombian Aedes aegypti Populations' has been provisionally accepted for publication in PLOS Neglected Tropical Diseases.

Best regards,

Jeffrey H Withey

Associate Editor

Tereza Magalhaes

Deputy Editor

---

## [Editor Report · Acceptance letter]

9 Dec 2021

Dear Dr Triana-Chávez,

We are delighted to inform you that your manuscript, "Molecular Surveillance of Resistance to Pyrethroids Insecticides in Colombian Aedes aegypti Populations," has been formally accepted for publication in PLOS Neglected Tropical Diseases.

Best regards,

Shaden Kamhawi

co-Editor-in-Chief

Paul Brindley

co-Editor-in-Chief
